# Stochastic Variational Deep Kernel Learning

**Andrew Gordon Wilson\***
Cornell University

**Zhiting Hu\***
CMU

**Ruslan Salakhutdinov**
CMU

**Eric P. Xing**
CMU

## Abstract

Deep kernel learning combines the non-parametric flexibility of kernel methods with the inductive biases of deep learning architectures. We propose a novel deep kernel learning model and stochastic variational inference procedure which generalizes deep kernel learning approaches to enable classification, multi-task learning, additive covariance structures, and stochastic gradient training. Specifically, we apply additive base kernels to subsets of output features from deep neural architectures, and jointly learn the parameters of the base kernels and deep network through a Gaussian process marginal likelihood objective. Within this framework, we derive an efficient form of stochastic variational inference which leverages local kernel interpolation, inducing points, and structure exploiting algebra. We show improved performance over stand alone deep networks, SVMs, and state of the art scalable Gaussian processes on several classification benchmarks, including an airline delay dataset containing 6 million training points, CIFAR, and ImageNet.

## 1 Introduction

Large datasets provide great opportunities to learn rich statistical representations, for accurate predictions and new scientific insights into our modeling problems. Gaussian processes are promising for large data problems, because they can grow their information capacity with the amount of available data, in combination with automatically calibrated model complexity [21, 25].

From a Gaussian process perspective, all of the statistical structure in data is learned through a kernel function. Popular kernel functions, such as the RBF kernel, provide smoothing and interpolation, but cannot learn representations necessary for long range extrapolation [22, 25]. With smoothing kernels, we can only use the information in a large dataset to learn about noise and length-scale hyperparameters, which tell us only how quickly correlations in our data vary with distance in the input space. If we learn a short length-scale hyperparameter, then by definition we will only make use of a small amount of training data near each testing point. If we learn a long length-scale, then we could subsample the data and make similar predictions.

Therefore to fully use the information in large datasets, we must build kernels with great representational power and useful learning biases, and scale these approaches without sacrificing this representational ability. Indeed many recent approaches have advocated building expressive kernel functions [e.g., 22, 9, 26, 25, 17, 31], and emerging research in this direction takes inspiration from deep learning models [e.g., 28, 5, 3]. However, the scalability, general applicability, and interpretability of such approaches remain a challenge. Recently, Wilson et al. [30] proposed simple and scalable deep kernels for single-output regression problems, with promising performance on many experiments. But their approach does not allow for stochastic training, multiple outputs, deep architectures with many output features, or classification. And it is on classification problems, in particular, where we often have high dimensional input vectors, with little intuition about how these vectors should correlate, and therefore most want to *learn* a flexible non-Euclidean similarity metric [1].

In this paper, we introduce inference procedures and propose a new deep kernel learning model which enables (1) classification and non-Gaussian likelihoods; (2) multi-task learning[1]; (3) stochastic gradient mini-batch training; (4) deep architectures with many output features; (5) additive covariance structures; and (5) greatly enhanced scalability.

We propose to use additive base kernels corresponding to Gaussian processes (GPs) applied to subsets of output features of a deep neural architecture. We then linearly mix these Gaussian processes, inducing correlations across multiple output variables. The result is a deep probabilistic neural network, with a hidden layer composed of additive sets of infinite basis functions, linearly mixed to produce correlated output variables. All parameters of the deep architecture and base kernels are *jointly learned* through a marginal likelihood objective, having integrated away all GPs. For scalability and non-Gaussian likelihoods, we derive stochastic variational inference (SVI) which leverages local kernel interpolation, inducing points, and structure exploiting algebra, and a hybrid sampling scheme, building on Wilson and Nickisch [27], Wilson et al. [29], Titsias [24], Hensman et al. [10], and Nickson et al. [18]. The resulting approach, SV-DKL, has a complexity of $\mathcal{O}(m^{1+1/D})$ for $m$ inducing points and $D$ input dimensions, versus the standard $\mathcal{O}(m^3)$ for efficient stochastic variational methods.

We achieve good predictive accuracy and scalability over a wide range of classification tasks, while retaining a straightforward, general purpose, and highly practical probabilistic non-parametric representation, with code available at `https://people.orie.cornell.edu/andrew/code`.

## 2   Background

Throughout this paper, we assume we have access to vectorial input-output pairs $\mathcal{D} = \{\mathbf{x}_i, \mathbf{y}_i\}$, where each $\mathbf{y}_i$ is related to $\mathbf{x}_i$ through a Gaussian process and observation model. For example, in regression, one could model $\mathbf{y}(\mathbf{x})|\mathbf{f}(\mathbf{x}) \sim \mathcal{N}(\mathbf{y}(\mathbf{x}); \mathbf{f}(\mathbf{x}), \sigma^2 I)$, where $\mathbf{f}(\mathbf{x})$ is a latent vector of independent Gaussian processes $\mathbf{f}_j \sim \mathcal{GP}(0, k_j)$, and $\sigma^2 I$ is a noise covariance matrix.

The computational bottleneck in working with Gaussian processes typically involves computing $(K_{X,X} + \sigma^2 I)^{-1}\mathbf{y}$ and $\log|K_{X,X}|$ over an $n \times n$ covariance matrix $K_{X,X}$ evaluated at $n$ training inputs $X$. Standard procedure is to compute the Cholesky decomposition of $K_{X,X}$, which incurs $\mathcal{O}(n^3)$ computations and $\mathcal{O}(n^2)$ storage, after which predictions cost $\mathcal{O}(n^2)$ per test point. Gaussian processes are thus typically limited to at most a few thousand training points. Many promising approaches to scalability have been explored, for example, involving randomized methods [20, 16, 31] , and low rank approximations [23, 19]. Wilson and Nickisch [27] recently introduced the KISS-GP approximate kernel matrix $\widetilde{K}_{X,X'} = M_X K_{Z,Z} M_{X'}^\top$, which admits fast computations, given the exact kernel matrix $K_{Z,Z}$ evaluated on a latent multidimensional lattice of $m$ inducing inputs $Z$, and $M_X$, a sparse interpolation matrix. Without requiring any grid structure in $X$, $K_{Z,Z}$ decomposes into a Kronecker product of Toeplitz matrices, which can be approximated by circulant matrices [29]. Exploiting such structure in combination with local kernel interpolation enables one to use many inducing points, resulting in near-exact accuracy in the kernel approximation, and $\mathcal{O}(n)$ inference. Unfortunately, this approach does not typically apply to $D > 5$ dimensional inputs [29].

Moreover, the Gaussian process marginal likelihood does not factorize, and thus stochastic gradient descent does not ordinarily apply. To address this issue, Hensman et al. [10] extended the variational approach from Titsias [24] and derived a stochastic variational GP posterior over inducing points for a regression model which does have the required factorization for stochastic gradient descent. Hensman et al. [12], Hensman et al. [11], and Dezfouli and Bonilla [6] further combine this with a sampling procedure for estimating non-conjugate expectations. These methods have $\mathcal{O}(m^3)$ sampling complexity which becomes prohibitive where many inducing points are desired for accurate approximation. Nickson et al. [18] consider Kronecker structure in the stochastic approximation of Hensman et al. [10] for regression, but do not leverage local kernel interpolation or sampling.

To address these limitations, we introduce a new deep kernel learning model for multi-task classification, mini-batch training, and scalable kernel interpolation which does not require low dimensional input spaces. In this paper, we view scalability and flexibility as two sides of one coin: we most want the flexible models on the largest datasets, which contain the necessary information to discover rich

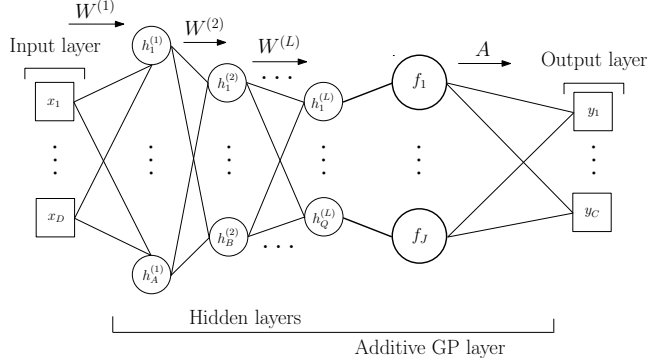

Figure 1: Deep Kernel Learning for Multidimensional Outputs. Multidimensional inputs $\mathbf{x} \in \mathbb{R}^D$ are mapped through a deep architecture, and then a series of additive Gaussian processes $f_1, \ldots, f_J$, with base kernels $k_1, \ldots, k_J$, are each applied to subsets of the network features $h_1^{(L)}, \ldots, h_Q^{(L)}$. The thick lines indicate a probabilistic mapping. The additive Gaussian processes are then linearly mixed by the matrix $A$ and mapped to output variables $y_1, \ldots, y_C$ (which are then correlated through $A$). All of the parameters of the deep network, base kernel, and mixing layer, $\boldsymbol{\gamma} = \{\mathbf{w}, \boldsymbol{\theta}, A\}$ are learned *jointly* through the (variational) marginal likelihood of our model, having integrated away all of the Gaussian processes. We can view the resulting model as a Gaussian process which uses an additive series of deep kernels with weight sharing.

statistical structure. We show that the resulting approach can learn very expressive and interpretable kernel functions on large classification datasets, containing millions of training points.

## 3   Deep Kernel Learning for Multi-task Classification

We propose a new deep kernel learning approach to account for classification and non-Gaussian likelihoods, multiple correlated outputs, additive covariances, and stochastic gradient training.

We propose to build a probabilistic deep network as follows: 1) a deep non-linear transformation $\mathbf{h}(\mathbf{x}, \mathbf{w})$, parametrized by weights $\mathbf{w}$, is applied to the observed input variable $\mathbf{x}$, to produce $Q$ features at the final layer $L$, $h_1^{(L)}, \ldots, h_Q^{(L)}$; 2) $J$ Gaussian processes, with base kernels $k_1, \ldots, k_J$, are applied to subsets of these features, corresponding to an additive GP model [e.g., 7]. The base kernels can thus act on relatively low dimensional inputs, where local kernel interpolation and learning biases such as similarities based on Euclidean distance are most natural; 3) these GPs are linearly mixed by a matrix $A \in \mathbb{R}^{C \times J}$, and transformed by an observation model, to produce the output variables $y_1, \ldots, y_C$. The mixing of these variables through $A$ produces *correlated* multiple outputs, a multi-task property which is uncommon in Gaussian processes or SVMs. The structure of this network is illustrated in Figure 1. Critically, all of the parameters in the model (including base kernel hyperparameters) are trained through optimizing a marginal likelihood, having integrated away the Gaussian processes, through the variational inference procedures described in section 4.

For classification, we consider a special case of this architecture. Let $C$ be the number of classes, and we have data $\{\mathbf{x}_i, \mathbf{y}_i\}_{i=1}^n$, where $\mathbf{y}_i \in \{0, 1\}^C$ is a one-shot encoding of the class label. We use the softmax observation model:

$$p(\mathbf{y}_i | \mathbf{f}_i, A) = \frac{\exp(A(\mathbf{f}_i)^\top \mathbf{y}_i)}{\sum_c \exp(A(\mathbf{f}_i)^\top \mathbf{e}_c)}, \tag{1}$$

where $\mathbf{f}_i \in \mathbb{R}^J$ is a vector of independent Gaussian processes followed by a linear mixing layer $A(\mathbf{f}_i) = A\mathbf{f}_i$; and $\mathbf{e}_c$ is the indicator vector with the $c$th element being 1 and the rest 0.

For the $j$th Gaussian process in the additive GP layer, let $\mathbf{f}_j = \{f_{ij}\}_{i=1}^n$ be the latent functions on the input data features. By introducing a set of latent inducing variables $\mathbf{u}_j$ indexed by $m$ inducing inputs $Z$, we can write [e.g., 19]

$$p(\mathbf{f}_j | \mathbf{u}_j) = \mathcal{N}(\mathbf{f}_j | K_{X,Z}^{(j)} K_{Z,Z}^{(j),-1} \mathbf{u}_j, \widetilde{K}^{(j)}), \quad \widetilde{K} = K_{X,X} - K_{X,Z} K_{Z,Z}^{-1} K_{Z,X}. \tag{2}$$

Substituting the local interpolation approximation $K_{X,X'} = M K_{Z,Z} M^\top$ of Wilson and Nickisch [27] into Eq. (2), we find $\widetilde{K}^{(j)} = 0$; it therefore follows that $\mathbf{f}_j = K_{X,Z} K_{Z,Z}^{-1} \mathbf{u} = M\mathbf{u}$. In section 4 we exploit this deterministic relationship between $\mathbf{f}$ and $\mathbf{u}$, governed by the sparse matrix $M$, to derive a particularly efficient stochastic variational inference procedure.

Eq. (1) and Eq. (2) together form the additive GP layer and the linear mixing layer of the proposed deep probabilistic network in Figure 1, with all parameters (including network weights) trained jointly through the Gaussian process marginal likelihood.

## 4   Structure Exploiting Stochastic Variational Inference

Exact inference and learning in Gaussian processes with a non-Gaussian likelihood is not analytically tractable. Variational inference is an appealing approximate technique due to its automatic regularization to avoid overfitting, and its ability to be used with stochastic gradient training, by providing a factorized approximation to the Gaussian process marginal likelihood. We develop our stochastic variational method equipped with a fast sampling scheme for tackling any intractable marginalization.

Let $\mathbf{u} = \{\mathbf{u}_j\}_{j=1}^J$ be the collection of the inducing variables of the $J$ additive GPs. We assume a variational posterior over the inducing variables $q(\mathbf{u})$. By Jensen's inequality we have

$$\log p(\mathbf{y}) \geq \mathbb{E}_{q(\mathbf{u})p(\mathbf{f}|\mathbf{u})}[\log p(\mathbf{y}|\mathbf{f})] - \mathrm{KL}[q(\mathbf{u})\|p(\mathbf{u})] \triangleq \mathcal{L}(q), \tag{3}$$

where we have omitted the mixing weights $A$ for clarity. The KL divergence term can be interpreted as a regularizer encouraging the approximate posterior $q(\mathbf{u})$ to be close to the prior $p(\mathbf{u})$. We aim at tightening the marginal likelihood lower bound $\mathcal{L}(q)$ which is equivalent to minimizing the KL divergence from $q$ to the true posterior.

Since the likelihood function typically factorizes over data instances: $p(\mathbf{y}|\mathbf{f}) = \prod_{i=1}^n p(\mathbf{y}_i|\mathbf{f}_i)$, we can optimize the lower bound with stochastic gradients. In particular, we specify $q(\mathbf{u}) = \prod_j \mathcal{N}(\mathbf{u}_j|\boldsymbol{\mu}_j, \mathbf{S}_j)$ for the independent GPs, and iteratively update the variational parameters $\{\boldsymbol{\mu}_j, \mathbf{S}_j\}_{j=1}^J$ and the kernel and deep network parameters using a noisy approximation of the gradient of the lower bound on minibatches of the full data. Henceforth we omit the index $j$ for clarity.

Unfortunately, for general non-Gaussian likelihoods the expectation in Eq (3) is usually intractable. We develop a sampling method for tackling this intractability which is highly efficient with structured reparameterization, local kernel interpolation, and structure exploiting algebra.

Using local kernel interpolation, the latent function $\mathbf{f}$ is expressed as a deterministic local interpolation of the inducing variables $\mathbf{u}$ (section 3). This result allows us to work around any difficult approximate posteriors on $\mathbf{f}$ which typically occur in variational approaches for GPs. Instead, our sampler only needs to account for the uncertainty on $\mathbf{u}$. The direct parameterization of $q(\mathbf{u})$ yields a straightforward and efficient sampling procedure. The latent function samples (indexed by $t$) are then computed directly through interpolation $\mathbf{f}^{(t)} = M\mathbf{u}^{(t)}$.

As opposed to conventional mean-field methods, which assume a diagonal variational covariance matrix, we use the Cholesky decomposition for reparameterizing $\mathbf{u}$ in order to preserve structures within the covariance. Specifically, we let $\mathbf{S} = \mathbf{L}^T\mathbf{L}$, resulting in the following sampling procedure:

$$\mathbf{u}^{(t)} = \boldsymbol{\mu} + \mathbf{L}\boldsymbol{\epsilon}^{(t)}; \quad \boldsymbol{\epsilon}^{(t)} \sim \mathcal{N}(\mathbf{0}, \mathbf{I}).$$

Each step of the above standard sampler has complexity of $\mathcal{O}(m^2)$, where $m$ is the number of inducing points. Due to the matrix vector product, this sampling procedure becomes prohibitive in the presence of many inducing points, which are required for accuracy on large datasets with multidimensional inputs – particularly if we have an expressive kernel function [27].

We scale up the sampler by leveraging the fact that the inducing points are placed on a grid (taking advantage of both Toeplitz and circulant structure), and additionally imposing a Kronecker decomposition on $\mathbf{L} = \bigotimes_{d=1}^D \mathbf{L}_d$, where $D$ is the input dimension of the base kernel. With the fast Kronecker matrix-vector products, we reduce the above sampling cost of $\mathcal{O}(m^2)$ to $\mathcal{O}(m^{1+1/D})$. Our approach thus greatly improves over previous stochastic variational methods which typically scale with $\mathcal{O}(m^3)$ complexity, as discussed shortly.

Note that the KL divergence term between the two Gaussians in Eq (3) has a closed form without the need for Monte Carlo estimation. Computing the KL term and its derivatives, with the Kronecker method, is $\mathcal{O}(Dm^{\frac{3}{D}})$. With $T$ samples of $\mathbf{u}$ and a minibatch of data points of size $B$, we can estimate the marginal likelihood lower bound as

$$\mathcal{L} \simeq \frac{N}{TB} \sum_{t=1}^T \sum_{i=1}^B \log p(\mathbf{y}_i|\mathbf{f}_i^{(t)}) - \mathrm{KL}[q(\mathbf{u})\|p(\mathbf{u})], \tag{4}$$

and the derivatives $\nabla\mathcal{L}$ w.r.t the model hyperparameters $\gamma$ and the variational parameters $\{\boldsymbol{\mu}, \{\mathbf{L}_d\}_{d=1}^D\}$ can be taken similarly. We provide the detailed derivation in the supplement.

Although a small body of pioneering work has developed stochastic variational methods for Gaussian processes, our approach distinctly provides the above representation-preserving variational approximation, and exploits algebraic structure for significant advantages in scalability and accuracy. In particular, a similar variational lower bound as in Eq (3) was proposed in [24, 10] for a sparse GP, which were extended to non-conjugate likelihoods, with the intractable integrals estimated using Gaussian quadrature as in the KLSP-GP [11] or univariate Gaussian samples as in the SAVI-GP [6]. Hensman et al. [12] estimates nonconjugate expectations with a hybrid Monte Carlo sampler (denoted as MC-GP). The computations in these approaches can be costly, with $\mathcal{O}(m^3)$ complexity, due to a complicated variational posterior over $\mathbf{f}$ as well as the expensive operations on the full inducing point matrix. In addition to its increased efficiency, our sampling scheme is much simpler, without introducing any additional tuning parameters. We empirically compare with these methods and show the practical significance of our algorithm in section 5.

Variational methods have also been used in GP regression for stochastic inference (e.g., [18, 10]), and some of the most recent work in this area applied variational auto-encoders [14] for coupled variational updates (aka back constraints) [4, 2]. We note that these techniques are orthogonal and complementary to our inference approach, and can be leveraged for further enhancements.

# 5 Experiments

We evaluate our proposed approach, stochastic variational deep kernel learning (SV-DKL), on a wide range of classification problems, including an airline delay task with over 5.9 million data points (section 5.1), a large and diverse collection of classification problems from the UCI repository (section 5.2), and image classification benchmarks (section 5.3). Empirical results demonstrate the practical significance of our approach, which provides consistent improvements over stand-alone DNNs, while preserving a GP representation, and dramatic improvements in speed and accuracy over modern state of the art GP models. We use classification accuracy when comparing to DNNs, because it is a standard for evaluating classification benchmarks with DNNs. However, we also compute the negative log probability (NLP) values (supplement), which show similar trends.

All experiments were performed on a Linux machine with eight 4.0GHz CPU cores, one Tesla K40c GPU, and 32GB RAM. We implemented deep neural networks with Caffe [13].

**Model Training**   For our deep kernel learning model, we used deep neural networks which produce $C$-dimensional top-level features. Here $C$ is the number of classes. We place a Gaussian process on each dimension of these features. We used RBF base kernels. The additive GP layer is then followed by a linear mixing layer $A \in \mathbb{R}^{C \times C}$. We initialized $A$ to be an identity matrix, and optimized in the joint learning procedure to recover cross-dimension correlations from data.

We first train a deep neural network using SGD with the softmax loss objective, and rectified linear activation functions. After the neural network has been pre-trained, we fit an additive KISS-GP layer, followed by a linear mixing layer, using the top-level features of the deep network as inputs. Using this pre-training *initialization*, our joint SV-DKL model of section 3 is then trained through the stochastic variational method of section 4 which jointly optimizes *all* the hyperparameters $\gamma$ of the deep kernel (including all network weights), as well as the variational parameters, by backpropagating derivatives through the proposed marginal likelihood lower bound of the additive Gaussian process in section 4. In all experiments, we use a relatively large mini-batch size (specified according to the full data size), enabled by the proposed structure exploiting variational inference procedures. We achieve good performance setting the number of samples $T = 1$ in Eq. 4 for expectation estimation in variational inference, which provides additional confirmation for a similar observation in [14].

## 5.1 Airline Delays

We first consider a large airline dataset consisting of flight arrival and departure details for all commercial flights within the US in 2008. The approximately 5.9 million records contain extensive information about the flights, including the delay in reaching the destination. Following [11], we consider the task of predicting whether a flight was subject to delay based on 8 features (e.g., distance to be covered, day of the week, etc).

**Classification accuracy**   Table 1 reports the classification accuracy of 1) KLSP-GP [11], a recent scalable variational GP classifier as discussed in section 4; 2) stand-alone deep neural network (DNN); 3) DNN+, a stand-alone DNN with an extra $Q \times c$ fully-connected hidden layer with $Q$, $c$ defined as in Figure 1; 4) DNN+GP which is a GP applied to a pre-trained DNN (with same architecture as in 2); and 5) our stochastic variational DKL method (SV-DKL) (same DNN architecture as in 2). For DNN, we used a fully-connected architecture with layers d-1000-1000-500-50-c.[2] The DNN component of the SV-DKL model has the exact same architecture. The SV-DKL joint training was conducted using a large minibatch size of 50,000 to reduce the variance of the stochastic gradient. We can use such a large minibatch in each iteration (which is daunting for regular GP even as a whole dataset) due to the efficiency of our inference strategy within each mini-batch, leveraging structure exploiting algebra.

From the table we see that SV-DKL outperforms both the alternative variational GP model (KLSP-GP) and the stand-alone deep network. DNN+GP outperforms stand-alone DNNs, showing the non-parametric flexibility of kernel methods. By combining KISS-GP with DNNs as part of a joint SV-DKL procedure, we obtain better results than DNN and DNN+GP. Besides, both the plain DNN and SV-DKL notably improve on stand-alone GPs, indicating a superior capacity of deep architectures to learn representations from large but finite training sets, despite the asymptotic approximation properties of Gaussian processes. By contrast, adding an extra hidden layer, as in DNN+, does not improve performance.

Figure 2(a) further studies how performance changes as data size increases. We observe that the proposed SV-DKL classifier trained on 1/50 of the data already can reach a competitive accuracy as compared to the KLSP-GP model trained on the full dataset. As the number of the training points increases, the SV-DKL and DNN models continue to improve. This experiment demonstrates the value of *expressive* kernel functions on large data problems, which can effectively capture the extra information available as seeing more training instances. Furthermore, SV-DKL consistently provides better performance over the plain DNN, through its non-parametric flexibility.

**Scalability**   We next measure the scalability of our variational DKL in terms of the number of inducing points $m$ in each GP. Figure 2(c) shows the runtimes in seconds, as a function of $m$, for evaluating the objective and any relevant derivatives. We compare our structure exploiting variational method with the scalable variational inference in KLSP-GP, and the MCMC-based variational method in MC-GP [12]. We see that our inference approach is far more efficient than previous scalable algorithms. Moreover, when the number of inducing points is not too large (e.g., $m = 70$), the added time for SV-DKL over DNN is reasonable (e.g., 0.39s vs. 0.27s), especially considering the gains in performance and expressive power. Figure 2(d) shows the runtime scaling of different variational methods as $m$ grows. We can see that the runtime of our approach increases only slowly in a wide range of $m$ ($< 2,000$), greatly enhancing the scalability over the other methods. This empirically validates the improved time complexity of our new inference method as presented in section 4.

We next investigate the total training time of the models. Table 1, right panel, lists the time cost of training KLSP-GP, DNN, and SV-DKL; and Figure 2(b) shows how the training time of SV-DKL and DNN changes as more training data is presented. We see that on the full dataset DKL, as a GP model, saves over 60% time as compared to the modern state of the art KLSP-GP, while at the same time achieving over an 18% improvement in predictive accuracy. Generally, the training time of SV-DKL increases slowly with growing data sizes, and has only modest additional overhead compared to stand-alone architectures, justified by improvements in performance, and the general benefits of a non-parametric probabilistic representation. Moreover, the DNN was fully trained on a GPU, while in SV-DKL the base kernel hyperparameters and variational parameters were optimized on a CPU. *Since most updates of the SV-DKL parameters are computed in matrix forms, we believe the already modest time gap between SV-DKL and DNNs can be almost entirely closed by deploying the whole SV-DKL model on GPUs.*

## 5.2   UCI Classification Tasks

The second evaluation of our proposed algorithm (SV-DKL) is conducted on a number of commonly used UCI classification tasks of varying sizes and properties. Table 1 (supplement) lists the classification accuracy of SVM, DNN, DNN+ (a stand-alone DNN with an extra $Q \times c$ fully-connected hidden layer with $Q$, $c$ defined as in Figure 1), DNN+GP (a GP trained on the top level features of a trained DNN without the extra hidden layer), and SV-DKL (same architecture as DNN).

Table 1: Classification accuracy and training time on the airline delay dataset, with $n$ data points, $d$ input dimensions, and $c$ classes. KLSP-GP is a stochastic variational GP classifier proposed in [11]. DNN+ is the DNN with an extra hidden layer. DNN+GP is a GP applied to fixed pre-trained output layer of the DNN (without the extra hidden layer). Following Hensman et al. [11], we selected a hold-out sets of 100,000 points uniformly at random, and the results of DNN and SV-DKL are averaged over 5 runs $\pm$ one standard deviation. Since the code of KLSP-GP is not publicly available we directly show the results from [11].

| Datasets | n | d | c | Accuracy | | | | | Total Training Time (h) | | |
|---|---|---|---|---|---|---|---|---|---|---|---|
| | | | | KLSP-GP [11] | DNN | DNN+ | DNN+GP | SV-DKL | KLSP-GP | DNN | SV-DKL |
| Airline | 5,934,530 | 8 | 2 | $\sim$0.675 | 0.773$\pm$0.001 | 0.722$\pm$0.002 | 0.7746$\pm$0.001 | **0.781$\pm$0.001** | $\sim$11 | 0.53 | 3.98 |

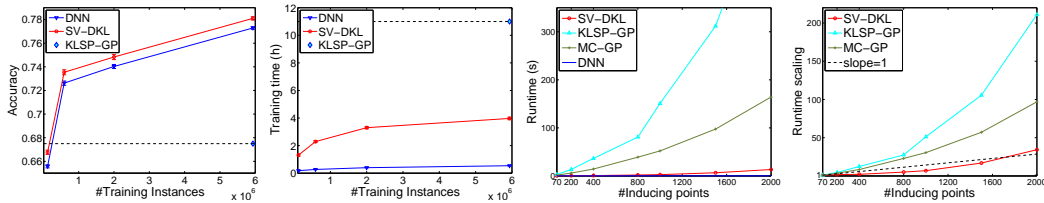

Figure 2: (a) Classification accuracy vs. the number of training points ($n$). We tested the deep models, DNN and SV-DKL, by training on 1/50, 1/10, 1/3, and the full dataset, respectively. For comparison, the cyan diamond and black dashed line show the accuracy level of KLSP-GP trained on the full data. (b) Training time vs. $n$. The cyan diamond and black dashed line show the training time of KLSP-GP on the full data. (c) Runtime vs. the number of inducing points ($m$) on airline task, by applying different variational methods for deep kernel learning. The minibatch size is fixed to 50,000. The runtime of the stand-alone DNN does not change as $m$ varies. (d) The scaling of runtime relative to the runtime of $m = 70$. The black dashed line indicates a slope of 1.

The plain DNN, which learns salient features effectively from raw data, gives notably higher accuracy compared to an SVM, the mostly widely used kernel method for classification problems. We see that the extra layer in DNN+GP can sometimes harm performance. By contrast, non-parametric flexibility of DNN+GP consistently improves upon DNN. And SV-DKL, by training a DNN through a GP marginal likelihood objective, consistently provides further enhancements (with particularly notable performance on the Connect4 and Covtype datasets).

## 5.3 Image Classification

We next evaluate the proposed scalable SV-DKL procedure for efficiently handling high-dimensional highly-structured image data. We used a minibatch size of 5,000 for stochastic gradient training of SV-DKL. Table 2 compares SV-DKL with the most recent scalable GP classifiers. Besides KLSP-GP, we also collected the results of the MC-GP [12] which uses a hybrid Monte Carlo sampler to tackle non-conjugate likelihoods, SAVI-GP [6] which approximates with a univariate Gaussian sampler, as well as the distributed GP latent variable model (denoted as D-GPLVM) [8]. We see that on the respective benchmark tasks, SV-DKL improves over all of the above scalable GP methods by a large margin. We note that these datasets are very challenging for conventional GP methods.

We further compare SV-DKL to stand-alone convolutional neural networks, and GPs applied to fixed pre-trained CNNs (CNN+GP). On the first three datasets in Table 2, we used the reference CNN models implemented in Caffe; and for the SVHN dataset, as no benchmark architecture is available, we used the CIFAR10 architecture which turned out to perform quite well. As we can see, the SV-DKL model outperforms CNNs and CNN+GP on all datasets. By contrast, the extra hidden $Q \times c$ hidden layer CNN+ does not consistently improve performance over CNN.

**ResNet Comparison**: Based on one of the best public implementations on Caffe, the ResNet-20 has 0.901 accuracy on CIFAR10, and SV-DKL (with this ResNet base architecture) improves to 0.910.

**ImageNet**: We randomly selected 20 categories of images with an AlexNet variant as the base NN [15], which has an accuracy of 0.6877, while SV-DKL achieves 0.7067 accuracy.

### 5.3.1 Interpretation

In Figure 3(a) we investigate the deep kernels learned on the MNIST dataset by randomly selecting 4 classes and visualizing the covariance matrices of respective dimensions. The covariance matrices are evaluated on the set of test inputs, sorted in terms of the labels of the input images. We see that the

Table 2: Classification accuracy on the image classification benchmarks. MNIST-Binary is the task to differentiate between odd and even digits on the MNIST dataset. We followed the standard training-test set partitioning of all these datasets. We have collected recently published results of a variety of scalable GPs. For CNNs, we used the respective benchmark architectures (or with slight adaptations) from Caffe. CNN+ is a stand-alone CNN with $Q \times c$ fully connected extra hidden layer. See the text for more details, including a comparison with ResNets on CIFAR10.

| Datasets | n | d | c | Accuracy | | | | | | | |
| --- | --- | --- | --- | --- | --- | --- | --- | --- | --- | --- | --- |
| | | | | MC-GP [12] | SAVI-GP [6] | D-GPLVM [8] | KLSP-GP [11] | CNN | CNN+ | CNN+GP | SV-DKL |
| MNIST-Binary | 60K | 28×28 | 2 | — | — | — | 0.978 | 0.9934 | 0.8838 | 0.9938 | **0.9940** |
| MNIST | 60K | 28×28 | 10 | 0.9804 | 0.9749 | 0.9405 | — | 0.9908 | 0.9909 | 0.9915 | **0.9920** |
| CIFAR10 | 50K | 3×32×32 | 10 | — | — | — | — | 0.7592 | 0.7618 | 0.7633 | **0.7704** |
| SVHN | 73K | 3×32×32 | 10 | — | — | — | — | 0.9214 | 0.9193 | 0.9221 | **0.9228** |

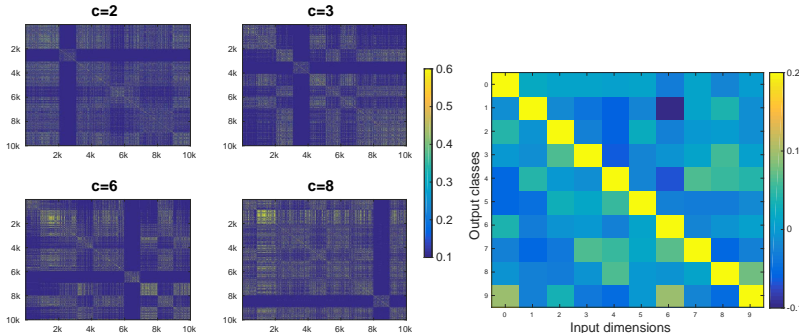

Figure 3: (a) The induced covariance matrices on classes 2, 3, 6, and 8, on test cases of the MNIST dataset ordered according to the labels. (b) The final mixing layer (i.e., matrix $A$) on MNIST digit recognition.

deep kernel on each dimension effectively discovers the correlations between the images within the corresponding class. For instance, in $c = 2$ the data points between 2k-3k (i.e., images of digit 2) are strongly correlated with each other, and carry little correlation with the rest of the images. Besides, we can also clearly observe that the rest of the data points also form multiple "blocks", rather than being crammed together without any structure. This validates that the DKL procedure and additive GPs do capture the correlations across different dimensions.

To further explore the learnt dependencies between the output classes and the additive GPs serving as the bases, we visualized the weights of the mixing layer ($A$) in Fig. 3(b), enabling the correlated multi-output (multi-task) nature of the model. Besides the expected high weights along the diagonal, we find that class 9 is also highly correlated with dimension 0 and 6, which is consistent with the visual similarity between digit "9" and "0"/"6". Overall, the ability to *interpret* the learned deep covariance matrix as discovering an expressive similarity metric across data instances is a distinctive feature of our approach.

## 6 Discussion

We introduced a scalable Gaussian process model which leverages deep learning, stochastic variational inference, structure exploiting algebra, and additive covariance structures. The resulting deep kernel learning approach, SV-DKL, allows for classification and non-Gaussian likelihoods, multi-task learning, and mini-batch training. SV-DKL achieves superior performance over alternative scalable GP models and stand-alone deep networks on many significant benchmarks.

Several fundamental themes emerge from the exposition: (1) kernel methods and deep learning approaches are *complementary*, and we can combine the advantages of each approach; (2) expressive kernel functions are particularly valuable on large datasets; (3) by viewing neural networks through the lens of *metric learning*, deep learning approaches become more interpretable.

Deep learning is able to obtain good predictive accuracy by automatically learning structure which would be difficult to a priori feature engineer into a model. In the future, we hope deep kernel learning approaches will be particularly helpful for interpreting these learned features, leading to new scientific insights into our modelling problems.

**Acknowledgements**: We thank NSF IIS-1563887, ONR N000141410684, N000141310721, N000141512791, and ADeLAIDE FA8750-16C-0130-001 grants.

## Footnotes

[1]We follow the GP convention where multi-task learning involves a function mapping a single input to multiple *correlated* output responses (class probabilities, regression responses, etc.). Unlike NNs which naturally have correlated outputs by sharing hidden basis functions (and multi-task can have a more specialized meaning), most GP models perform multiple binary classification, ignoring correlations between output classes. Even applying a GP to NN features for deep kernel learning does not naturally produce multiple correlated outputs.

[2]We obtained similar results with other DNN architectures (e.g., d-1000-1000-500-50-20-c).

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
