[Supplementary Material]

# Supplementary Material: Stochastic Variational Deep Kernel Learning

**Andrew Gordon Wilson***
Cornell University

**Zhiting Hu***
CMU

**Ruslan Salakhutdinov**
CMU

**Eric P. Xing**
CMU

Please see `https://people.orie.cornell.edu/andrew/code` for new results, updates, and code.

## 1 UCI Classification Results

The second evaluation of our proposed algorithm (SV-DKL) is conducted on a number of commonly used UCI classification tasks of varying sizes and properties. Table 1 lists the classification accuracy of SVM, DNN, DNN+ (a stand-alone DNN with an extra $Q \times c$ fully-connected hidden layer with $Q$, $c$ defined as in Figure 1 of the main text, DNN+GP (a GP trained on the top level features of a trained DNN without the extra hidden layer), and SV-DKL (same architecture as DNN).

The plain DNN, which learns salient features effectively from raw data, gives notably higher accuracy compared to an SVM, the mostly widely used kernel method for classification problems. We see that the extra layer in DNN+GP can sometimes harm performance. By contrast, non-parametric flexibility of DNN+GP consistently improves upon DNN. And SV-DKL, by training a DNN through a GP marginal likelihood objective, consistently provides further enhancements (with particularly notable performance on the Connect4 and Covtype datasets).

Table 1: Classification accuracy on the UCI datasets. We report the average accuracy $\pm$ one standard deviation using 5-fold cross validation. To compare with an SVM, we used the popular libsvm [1] toolbox. RBF kernel was used in SVM, and optimal hyper-parameters are selected automatically using the built-in functionality. On each dataset we used a fully-connected DNN which has the same architecture as in the airline delay task, except for DNN+ which has an additional hidden layer.

| Datasets | n | d | c | Accuracy | | | | |
|---|---|---|---|---|---|---|---|---|
| | | | | SVM | DNN | DNN+ | DNN+GP | SV-DKL |
| Adult | 48,842 | 14 | 2 | 0.849±0.001 | 0.852±0.001 | 0.845±0.001 | 0.853±0.001 | **0.857±0.001** |
| Connect4 | 67,557 | 42 | 3 | 0.773±0.002 | 0.805±0.005 | 0.804±0.009 | 0.811±0.005 | **0.841±0.006** |
| Diabetes | 101,766 | 49 | 25 | 0.869±0.000 | 0.889±0.001 | 0.890±0.001 | 0.891±0.001 | **0.896±0.001** |
| Covtype | 581,012 | 54 | 7 | 0.796±0.006 | 0.824±0.004 | 0.811±0.002 | 0.827±0.006 | **0.860±0.006** |

## 2 Negative Log Probability (NLP) Results

Tables 2, 3, and 4 show the negative log probability values on different tasks. Generally we observed similar trends as from the classification accuracy results.

Table 3: Negative log probability results on the UCI datasets, with the same experimental setting as in section 5.2.

| Datasets | n | d | c | NLP | | |
|---|---|---|---|---|---|---|
| | | | | DNN | DNN+GP | SV-DKL |
| Adult | 48,842 | 14 | 2 | 0.316±0.003 | 0.314±0.003 | **0.312±0.003** |
| Connect4 | 67,557 | 42 | 3 | 0.495±0.003 | 0.478±0.003 | **0.449±0.002** |
| Diabetes | 101,766 | 49 | 25 | 0.404±0.001 | 0.396±0.002 | **0.385±0.002** |
| Covtype | 581,012 | 54 | 7 | 0.435±0.004 | 0.429±0.005 | **0.365±0.004** |

Table 2: Negative log probability results on the airline delay dataset, with the same experimental setting as in section 5.1.

| Datasets | n | d | c | NLP | | | |
|---|---|---|---|---|---|---|---|
| | | | | KLSP-GP | DNN | DNN+GP | SV-DKL |
| Airline | 5,934,530 | 8 | 2 | ∼0.61 | 0.474±0.001 | 0.473±0.001 | **0.461±0.001** |

Table 4: Negative log probability results on the image classification benchmarks, with the same experimental setting as in section 5.3.

| Datasets | n | d | c | NLP | | | | |
|---|---|---|---|---|---|---|---|---|
| | | | | MC-GP | KLSP-GP | CNN | CNN+GP | SV-DKL |
| MNIST-Binary | 60K | 28×28 | 2 | — | 0.069 | 0.020 | 0.019 | **0.018** |
| MNIST | 60K | 28×28 | 10 | 0.064 | — | **0.028** | **0.028** | **0.028** |
| CIFAR10 | 50K | 3×32×32 | 10 | — | — | 0.738 | 0.738 | **0.734** |
| SVHN | 73K | 3×32×32 | 10 | — | — | 0.309 | 0.310 | **0.308** |

# 3 Stochastic Variational Inference for Deep Kernel Learning Classification

Recall the SV-DKL classification model

$$
\begin{aligned}
p(\mathbf{y}_i|\mathbf{f}_i, \mathbf{A}) &= \frac{\exp(\mathbf{a}(\mathbf{f}_i)^T \mathbf{y}_i)}{\sum_c \exp(\mathbf{a}(\mathbf{f}_i)^T \mathbf{e}_c)} \\
p(\mathbf{f}_j|\mathbf{u}_j) &= M^{(j)} \mathbf{u}_j \\
p(\mathbf{u}_j) &= \mathcal{N}(\mathbf{u}_j|\mathbf{0}, K_{Z,Z}^{(j)}),
\end{aligned}
\tag{1}
$$

Let $\mathbf{u} = \{\mathbf{u}_j\}_{j=1}^J$. We assume a variational posterior over the inducing variables

$$
q(\mathbf{u}) = \prod_j \mathcal{N}(\mathbf{u}_j|\boldsymbol{\mu}_j, \mathbf{S}_j)
\tag{2}
$$

By Jensen's inequality we have

$$
\begin{aligned}
\log p(\mathbf{y}) &\geq \mathbb{E}_{q(\mathbf{u})p(\mathbf{f}|\mathbf{u})}[\log p(\mathbf{y}|\mathbf{f})] - \mathrm{KL}[q(\mathbf{u})\|p(\mathbf{u})] \\
&\triangleq \mathcal{L}(q),
\end{aligned}
\tag{3}
$$

In the following we omit the GP index $j$ when there is no ambiguity. Due to the deterministic mapping, we can obtain latent function samples from the samples of $\mathbf{u}$:

$$
\mathbf{f}^{(t)} = M\mathbf{u}^{(t)}.
\tag{4}
$$

To sample from $q(\mathbf{u})$, we use the Cholesky decomposition for reparameterizing $\mathbf{u}$ in order to preserve structures within the covariance. Specifically, we let $\mathbf{S} = \mathbf{L}^T \mathbf{L}$. This results in the following sampling procedure for $\mathbf{u}$:

$$
\mathbf{u}^{(t)} = \boldsymbol{\mu} + \mathbf{L}\boldsymbol{\epsilon}^{(t)}; \quad \boldsymbol{\epsilon}^{(t)} \sim \mathcal{N}(\mathbf{0}, \mathbf{I}).
$$

We further scale up the sampler by leveraging the fact that the inducing points are placed on a grid, and imposing Kronecker decomposition on $\mathbf{L} = \bigotimes_{d=1}^{D} \mathbf{L}_d$, where $D$ is the input dimension of the base kernel. With the fast Kronecker matrix-vector products, the sampling cost is $\mathcal{O}(m^{1+1/D})$. Note that

$$\mathbf{S} = \left(\bigotimes \mathbf{L}_d\right)^T \left(\bigotimes \mathbf{L}_d\right) = \bigotimes_{d=1}^{D} \mathbf{L}_d^T \mathbf{L}_d := \bigotimes_{d=1}^{D} \mathbf{S}_d$$

With the samples, then for any $h(\mathbf{u})$, we have

$$
\begin{aligned}
\mathbb{E}_{q(\mathbf{u})}[h(\mathbf{u})] &\simeq \frac{1}{T} \sum_{t=1}^{T} h(\mathbf{u}^{(t)}) \\
&= \frac{1}{T} \sum_{l=1}^{T} h(\boldsymbol{\mu} + \mathbf{L}\boldsymbol{\epsilon}^{(t)}) \\
&\simeq E_{p(\boldsymbol{\epsilon})}[h(\boldsymbol{\mu} + \mathbf{L}\boldsymbol{\epsilon})]
\end{aligned}
\tag{5}
$$

Next we give the derivation of the objective lower bound and its derivatives in detail. In the following we denote $K := K_{Z,Z}$ for clarity.

**Computation of the marginal likelihood lower bound**  The expectation term of objective lower bound Eq (3) can be computed straightforwardly following Eq (5). The KL term has a closed form (we omit the GP index $j$):

$$\mathrm{KL}(q(\mathbf{u}) \| p(\mathbf{u})) = \frac{1}{2} \left\{ \log |K| - \log |\mathbf{S}| - D + tr(K^{-1}\mathbf{S}) + \boldsymbol{\mu}^T K^{-1} \boldsymbol{\mu} \right\}. \tag{6}$$

With the Kronecker product representation of the covariance matrices, all the above matrix operations can be conducted efficiently:

$$
\begin{aligned}
\log \det \mathbf{S} &= \log \prod_{d=1}^{D} \det(\mathbf{L}_d \mathbf{L}_d^T)^{\mathrm{rank}_d} \\
&= 2 \sum_{d=1}^{D} \mathrm{rank}_d \sum_{p=1}^{m_d} \log \mathbf{L}_{d,pp} \\
tr(K^{-1}\mathbf{S}) &= \prod_{d=1}^{D} tr(K_d^{-1} S_d),
\end{aligned}
\tag{7}
$$

where $m_d$ is the number of inducing points in dimension $d$ (we have $m = \prod_{d=1}^{D} m_d$); $\mathrm{rank}_d = \prod_{d' \neq d} \mathrm{rank}(\mathbf{S}_{d'})$; and $K = \bigotimes_{d=1}^{D} K_d$.

**Derivatives w.r.t the base kernel hyperparameters**  Note that the base kernel hyperparameters $\theta$ are only involved in the KL term of Eq (3). The derivative is

$$
\begin{aligned}
\frac{\partial \mathcal{L}}{\partial \theta} &= \frac{\partial \mathrm{KL}(q\|p)}{\partial \theta} \\
&= \frac{1}{2} \left\{ tr(K^{-1} \frac{\partial K}{\partial \theta}) + tr(\frac{\partial K^{-1}}{\partial \theta} S) - \boldsymbol{\mu}^T K^{-1} \frac{\partial K}{\partial \theta} K^{-1} \boldsymbol{\mu} \right\} \\
&= \frac{1}{2} \left\{ tr(K^{-1} \frac{\partial K}{\partial \theta}) - tr(K^{-1} \frac{\partial K}{\partial \theta} K^{-1} S) - \boldsymbol{\mu}^T K^{-1} \frac{\partial K}{\partial \theta} K^{-1} \boldsymbol{\mu} \right\}
\end{aligned}
\tag{8}
$$

Note that the matrix inversions and traces can be computed efficiently by leveraging the Kronecker product as in Eq (7).

**Derivatives w.r.t other model parameters**  Other model parameters, including the deep network weights and the top-layer mixing weights, are only involved in the likelihood expectation term in Eq (3), and can be computed conveniently by following Eq (5) with $h(\cdot)$ replaced by the respective derivatives of the softmax likelihood in Eq (1).

**Derivatives w.r.t the variational parameters** We only show the derivatives w.r.t the variational covariance parameters $\mathbf{L}$. The derivatives w.r.t the variational means $\boldsymbol{\mu}$ can be derived similarly.

(1) The derivative of the softmax expectation term of input $i$ w.r.t the $(p,q)$-th element of $\mathbf{L}_d^{(j)}$, denoted as $\lambda$ for clarity, is given by

$$\nabla_\lambda \log p(\mathbf{y}_i|\mathbf{f}_i) = \mathbb{E}_{p(\boldsymbol{\epsilon})}\left[\left(\sum_c \mathbf{1}(y_{ic}=1)A_{cj} - \frac{\exp(\mathbf{a}(\mathbf{f}_i)^T\mathbf{y}_i)}{\sum_c \exp(\mathbf{a}(\mathbf{f}_i)^T\mathbf{e}_c)}A_{cj}\right)M_{i\cdot}^{(j)}\nabla_\lambda L^{(j)}\boldsymbol{\epsilon}\right],$$

where $M_{i\cdot}^{(j)}$ is the $i$th row of the interpolation matrix (i.e., the interpolation vector of input $i$); and $\nabla_\lambda L^{(j)} = L_1^{(j)} \otimes \cdots \otimes \nabla_\lambda L_d^{(j)} \otimes \cdots \otimes L_D^{(j)}$. Note that for $D=1$, we can directly write down the derivatives w.r.t the whole matrix $\mathbf{L}^{(j)}$ which is efficient for computing:

$$\nabla_{L^{(j)}} \log p(\mathbf{y}_i|\mathbf{f}_i) = \mathbb{E}_{p(\boldsymbol{\epsilon})}\left[\left(\sum_c \mathbf{1}(y_{ic}=1)A_{cj} - \frac{\exp(\mathbf{a}(\mathbf{f}_i)^T\mathbf{y}_i)}{\sum_c \exp(\mathbf{a}(\mathbf{f}_i)^T\mathbf{e}_c)}A_{cj}\right)(\boldsymbol{\epsilon}M_{i\cdot}^{(j)})^T\right].$$

(2) The derivative of the KL term is (index $j$ omitted):

$$\nabla_\lambda \mathrm{KL}[q(\mathbf{u})\|p(\mathbf{u})] = \frac{1}{2}\frac{\partial - \log|S| + tr(K^{-1}S)}{\partial\lambda}$$
$$= -(L_d^{-1})_{pq} + tr(K_1^{-1}S_1)\cdots tr(K_d^{-1}L_d\nabla_\lambda L_d^T)\cdots tr(K_D^{-1}S_D),$$

where $tr(K_d^{-1}L_d\nabla_\lambda L_d^T) = (K_{d\cdot}^{-1}L_d)_{pq}$. Note that the Kronecker factor matrices are small (with size $m_d \times m_d$) and thus the above computations are fast.

## References

[1] C.-C. Chang and C.-J. Lin. Libsvm: a library for support vector machines. *ACM Transactions on Intelligent Systems and Technology (TIST)*, 2(3):27, 2011.