[Reviews · NeurIPS 2016]

Reviewer 1

Summary

This paper extend the "Deep Kernel Learning" idea for classification problems.

Qualitative Assessment

1. In line 39, you mention that your proposed method enables features like "multi-task learning", but I do not see any single experiment/discussion about it. Similarly, I do not follow what do you mean by "deep architectures with many output features". 2. I wonder how does your model perform if you only train the GP architecture of the networks, i.e. keep the pre-trained weights fixed. This way, the depth part of your model just provide better representation for GP in the back-end and you can make fair comparisons between GP, simple logistic regression and other classification methods. 3. How do you compare your proposed loss function in Eq 3 and 4 with cross entropy loss? and do you expect this loss integrated with your deep kernel model is a better loss function than cross entropy for classification tasks? If yes, why? by "better", I mean it leads better classification accuracy. 3. Significance of results: I wonder if the gains reported in Table 3 are significant? 4. Since you are proposing kind of a new classification method, why not reporting results on the state-of-the-are classification tasks? MNIST is too simple example in my opinion and there have been higher numbers in this task compare to the accuracies reported in Table 3.

Confidence in this Review

3-Expert (read the paper in detail, know the area, quite certain of my opinion)


Reviewer 2

Summary

Authors proposed a new deep kernel learning model for classification by embedding an additive GP layer into the neural network (before the output layer). Authors also proposed stochastic variational inference procedures to jointly optimize all parameters of the new model.

Qualitative Assessment

This paper is well written. It is reasonable and meaningful to build kernel learning models based on neural networks. The proposed deep kernel learning model and inference procedure are novel. However, the experiments are not enough to verify the significance of the proposed methods. 1) Authors claimed that the proposed model also enables multi-task learning, but experiments only involved classification. 2) Experiments were not conducted on state-of-the-art neural networks. The proposed model is actually based on the success of neural networks, and it is expected to perform better than traditional GPs. However, it is necessary to verify its usefulness in practice when building based on state-of-the-art neural networks, rather than only a simple neural network. E.g., I wonder how the proposed model performs if the ResNet-20, ResNet-56, or ResNet-110 model was adopted as the DNN model in the CIFAR10 dataset. 3) The proposed model can be considered to embed a GP layer to a neural network. I would like to know the performance of another DNN model with the GP layer is replaced by a DNN layer. This can verify whether the performance improvement of SV-DKL over DNN is from the GP layer or just increasing the "depth" of the DNN.

Confidence in this Review

1-Less confident (might not have understood significant parts)


Reviewer 3

Summary

The author produces an architecture containing a deep neural network with a final layer of Gaussian Processes. An approach is proposed to jointly train the full architecture with stochastic gradients, where GP gradients are obtained from a variational inference bound. Although this is not novel in itself, the authors propose a sampling scheme to provide scalable approximation to the posterior expectation and competitive results are established on a number of standard classification benchmarks (Airline, UCI, MNIST, CIFAR).

Qualitative Assessment

I think the paper is well written and provides substantial improvements in the intersection of kernel-based machine learning and deep learning. Although these architectures are complex and slow in comparison to vanilla DNNs, the paper's sampling scheme is not too complex and we see significant speedups over KLSP-GP which is encouraging. I feel confident this architecture will be used by researchers and practitioners, as deep kernel learning matures and believe it should be admitted to NIPS.

Confidence in this Review

1-Less confident (might not have understood significant parts)


Reviewer 4

Summary

This paper presents a model (SV-DKL) that use deep neural networks as feature extractors and apply additive base kernels to the output features. The deep neural networks are pre-trained followed by a training procedure of the additive base kernel and neural network parameters in cohesion. The parameters are optimized by variational inference and the learning has intriguing runtime properties. The model shows promising results compared to SVM and deep neural networks with and without GP.

Qualitative Assessment

This is a well-written paper with only a small number of grammatical and presentation errors, e.g. there is no numbering (a, b, c, d) in figure 2. The theoretical basis for this work is sound and explained well. However, the experimental section lacks some detail on the training of DNNs and the pre-training. From the level of detail given it will be hard to reproduce the results. Furthermore the results are not too convincing. The authors claim that "flexible deep kernel GPs achieve superior performance" to DNNs and SVMs. The difference in the classification results doesn't seem as convincing. It would be nice to have a more detailed discussion on whether the comparison is fair. Currently the DNNs in the stand-alone model and the DNN+GP/SV-DKL are all parameterized equally, meaning that the GP and SV-DKL additions both have more parameters. Furthermore the DNN+GP addition is trained independently, thus it will not have the same flexibility as the training of SV-DKL. Would the DNN produce better results if it were just defined with more parameters? As stated earlier it is not clear how the DNN is trained, but it would be beneficial to the training of the DNN with regularization (e.g. dropout); hence the KL term in the ELBO will work as a regularizer for the SV-DKL. In conclusion, this is a well-written paper, but it needs to argue a better case in order to be a convincing novel contribution.

Confidence in this Review

2-Confident (read it all; understood it all reasonably well)


Reviewer 5

Summary

The paper reads very well with clear motivation of incorporating kernel learning to deep networks. The authors do a really good job of explaining deep kernel learning for multi-task classification with the example of figure 1. Kernel methods and SVMs dominated various learning tasks and bringing in the kernel methods to deep learning is an interesting direction. While I do not know the motivation behind the two experiments chosen to compare the methods the empirical results show improvements in accuracy and training time. The only doubt is the time taken to train the new framework over vanilla DNNs as compared to the accuracy improvements. From the table 1, it seems it's 8x slower to train compared to DNNs.

Qualitative Assessment

It would be great to get a comment on speed of training and utility of it. Also, CIFAR 10 has saturated in numbers. It would be great to see the value of this model on tiny imagenet if not the whole imagenet to get a good sense of how much the non parametric aspect of the model is helping.

Confidence in this Review

1-Less confident (might not have understood significant parts)